# Short-lived metal-centered excited state initiates iron-methionine photodissociation in ferrous cytochrome c

Marco E. Reinhard[1,2], Michael W. Mara[3,5], Thomas Kroll[2], Hyeongtaek Lim [3], Ryan G. Hadt [3,6], Roberto Alonso-Mori [4], Matthieu Chollet[4], James M. Glownia[4], Silke Nelson[4], Dimosthenis Sokaras[2], Kristjan Kunnus[1], Tim Brandt van Driel[4], Robert W. Hartsock[1], Kasper S. Kjaer[1], Clemens Weninger[4], Elisa Biasin [1], Leland B. Gee[3], Keith O. Hodgson[2,3], Britt Hedman[2], Uwe Bergmann [1], Edward I. Solomon [2,3✉] & Kelly J. Gaffney [1,2✉]

The dynamics of photodissociation and recombination in heme proteins represent an archetypical photochemical reaction widely used to understand the interplay between chemical dynamics and reaction environment. We report a study of the photodissociation mechanism for the Fe(II)-S bond between the heme iron and methionine sulfur of ferrous cytochrome c. This bond dissociation is an essential step in the conversion of cytochrome c from an electron transfer protein to a peroxidase enzyme. We use ultrafast X-ray solution scattering to follow the dynamics of Fe(II)-S bond dissociation and 1s3p (Kβ) X-ray emission spectroscopy to follow the dynamics of the iron charge and spin multiplicity during bond dissociation. From these measurements, we conclude that the formation of a triplet metal-centered excited state with anti-bonding Fe(II)-S interactions triggers the bond dissociation and precedes the formation of the metastable Fe high-spin quintet state.

[1] PULSE Institute, SLAC National Accelerator Laboratory, Stanford University, Stanford, CA, USA. [2] Stanford Synchrotron Radiation Lightsource, SLAC National Accelerator Laboratory, Stanford University, Menlo Park, CA, USA. [3] Department of Chemistry, Stanford University, Stanford, CA, USA. [4] Linac Coherent Light Source, SLAC National Accelerator Laboratory, Stanford University, Menlo Park, CA, USA. [5] Present address: Department of Chemistry, Northwestern University, Evanston, IL, USA. [6] Present address: Division of Chemistry and Chemical Engineering, Arthur Amos Noyes Laboratory of Chemical Physics, California Institute of Technology, Pasadena, CA, USA. ✉email: edward.solomon@stanford.edu; kgaffney@slac.stanford.edu

Optogenetics and bioimaging applications have enhanced the significance of photochemical manipulation of proteins[1–4]. The potential significance of photochemical dynamics for cytochrome $c$ (cyt $c$) has been enhanced by the discovery that changes in axial ligand coordination are necessary to convert cyt $c$ to a peroxidase enzyme involved in apoptosis[5,6]. Horse heart cyt $c$ (Fig. 1a) consists of a single polypeptide chain with 104 amino acid residues where the iron porphyrin cofactor is transaxially ligated to histidine (His18) and methionine (Met80) residues of the single polypeptide. For ferrous cyt $c$, excitation of the heme $^1\pi-\pi^*$ electronic excited state (ES) (Fig. 1b) leads to dissociation of the heme-Met80 Fe(II)-S bond[7,8], which is one of the critical structural changes needed to transform cyt $c$ from an electron transfer protein into a peroxidase enzyme.

Understanding heme axial ligand dissociation has been a long-standing challenge. While the ultrafast nature of ligand dissociation has been robustly confirmed by ultrafast vibrational spectroscopies[7,9,10], the electronic ES that initiates the dissociation has not been clearly identified. For these heme proteins, the light absorption generates a $^1\pi-\pi^*$ excitation of the porphyrin ring. This excitation does not directly trigger axial ligand dissociation, which requires ES relaxation from the porphyrin to the Fe. For CO hemoglobin, ultrafast changes in the UV-visible spectrum have been interpreted to result from a transition from the $^1\pi-\pi^*$ state to a metal-to-ligand charge transfer (MLCT) state[11]. The MLCT promotes a $d_\pi$ ($d_{xz}$, $d_{yz}$) electron into the $\pi$ orbital vacated by light absorption, thus weakening the Fe–CO backbonding and initiating the Fe–CO dissociation. Such a mechanism appears less viable for cyt $c$ Fe(II)–S dissociation, since this bond lacks $\pi$ character[12]. Chergui and co-workers concluded from ultrafast spectroscopy measurements that the excited electron in the $\pi^*$ orbital transfers to the metal $d_{z^2}$ orbital, a ligand-to-metal charge transfer (LMCT) state[13,14]. The Fe–S antibonding character of this orbital provides a clear mechanism

for bond dissociation, but the energy of the $d_{z^2}$ orbital exceeds that of the $\pi^*$ orbital making this transition energetically infeasible[15,16].

The absence of clear optical signatures for metal-centered ES and methods capable of correlating electronic ES populations with the dynamics of Fe-ligand bond expansion have inhibited the experimental characterization of the photodissociation mechanisms of heme proteins. In a prior study of ferrous cyt $c$[8], we used the Fe K-edge X-ray absorption near-edge structure (XANES)[17–19] spectrum to characterize the structure around Fe in photoexcited heme confirming the dissociation of Met80, and $1s3p$ X-ray emission spectroscopy (Kβ XES) to confirm the high-spin quintet state of the resulting five-coordinate Fe(II)[8].

In the present study, we use ultrafast X-ray spectroscopy and X-ray solution scattering (XSS) to simultaneously track electronic and nuclear structure changes with femtosecond resolution to characterize the photodissociation mechanism[20–22]. We use ultrafast XSS[20,23] to track the dynamics of Fe(II)–S bond dissociation, and Kβ XES to correlate these dynamics with the ultrafast changes in the Fe charge and spin state (see Fig. 1c for the experimental setup)[20].

## Results

**Ultrafast Kβ X-ray emission spectroscopy.** The Kβ XES spectrum of $3d$ transition metal ions is sensitive to the effective $3d$ spin moment due to the strong exchange interaction between the unpaired $3d$ electrons and the one unpaired $3p$ electron in the final state created by the X-ray emission process (Fig. 2a)[24,25]. These attributes have been used to follow the femtosecond dynamics of charge transfer and intersystem crossing in transition metal complexes[20].

This Kβ XES study extends the work of Mara et al.[8] by investigating the potential role of electronic ES populated between

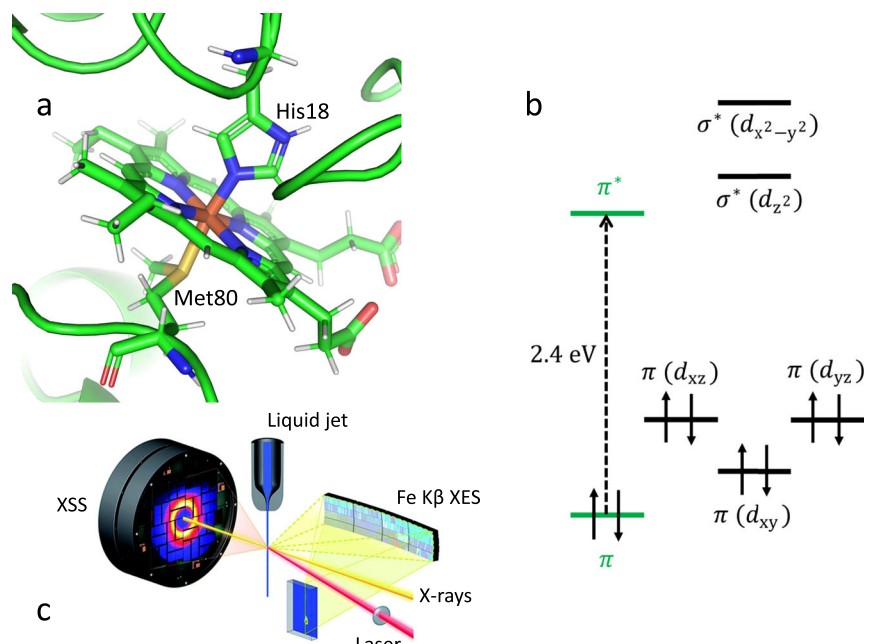

**Fig. 1 Photoinduced dynamics of ferrous cytochrome $c$ observed with femtosecond X-ray emission and scattering. a** Heme environment of reduced horse heart cytochrome $c$ (cyt $c$) with Met80 and His18 axial ligands. **b** Ferrous heme ground state electronic configuration of cyt $c$. The dashed arrow indicates the photoexcitation process. **c** Scheme of the experimental setup (adapted from Kjaer et al.[20]—published by The Royal Society of Chemistry). The X-ray pulses probe laser-induced changes from the cyt $c$ liquid jet sample. 2D images of the Fe Kβ X-ray emission spectra and the X-ray scattering in the forward direction are simultaneously read out shot-by-shot. The Fe Kβ X-ray emission signal is collected using a high-energy resolution X-ray emission spectrometer based on the von Hamos geometry.

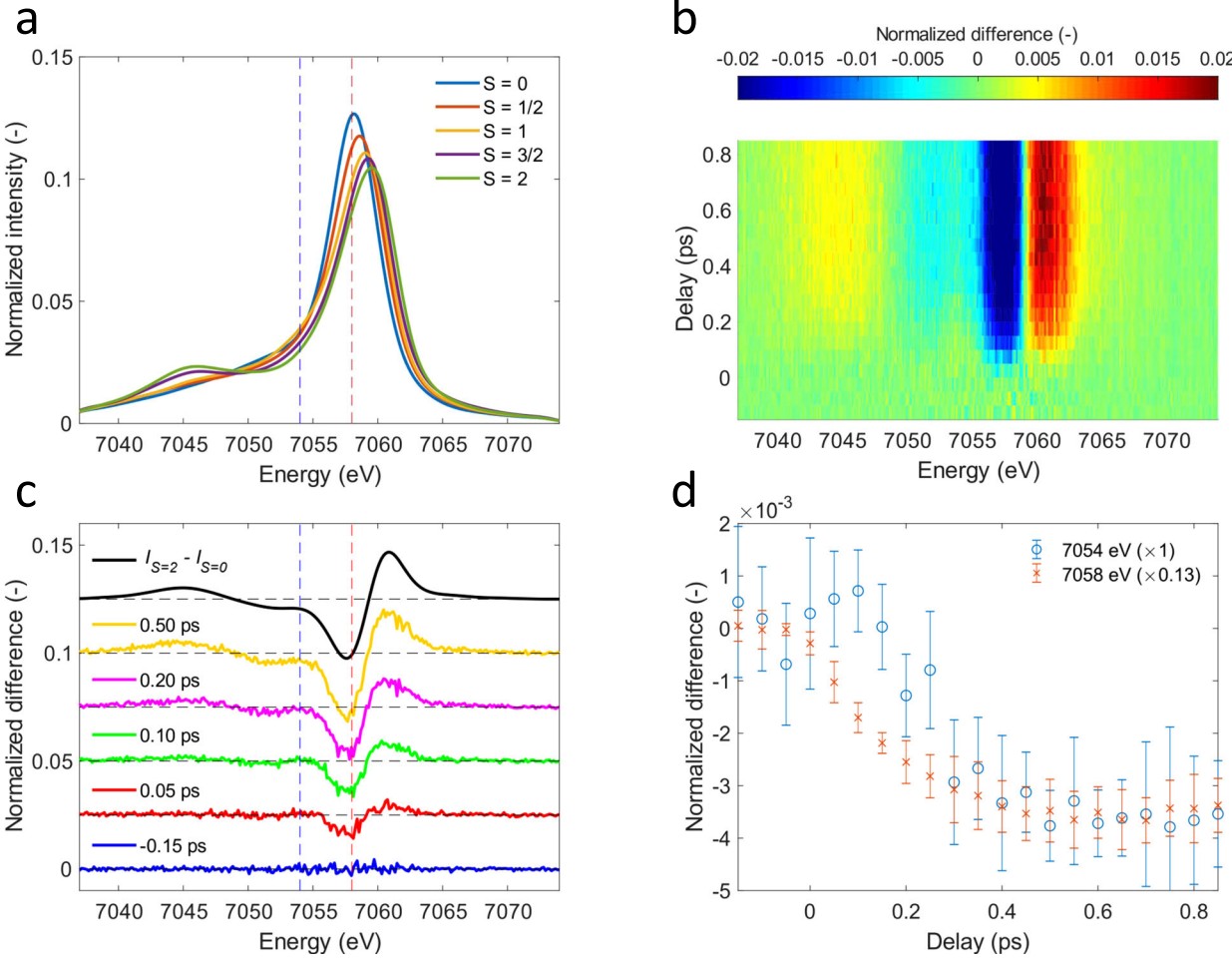

**Fig. 2 Time evolution of Fe Kβ X-ray emission difference spectra. a** Area-normalized Kβ X-ray emission spectroscopy (XES) references from Kjaer et al.[20]. Singlet ([Fe(2,2′-bipyridine)$_3$]$^{2+}$, blue), doublet ([Fe(2,2′-bipyridine)$_3$]$^{3+}$, red), triplet (Fe(II)phthalocyanine, orange), quartet (Fe(III) phthalocyanine chloride, purple), quintet ([Fe(phenanthroline)$_2$(NCS)$_2$], green)-. **b** Two-dimensional map of cytochrome *c* Kβ XES difference signal. **c** Difference spectra at various time delays. **d** Time-dependence for Kβ X-ray emission energies indicated by dashed vertical lines in **c**. Error bars reflect the standard deviation of the signal within a range of 7 detector pixels around these energies.

the optically generated $^1\pi-\pi^*$ state and the quintet metal-centered ($^5$MC) state observed for time delays beyond 600 fs. Given the minimal involvement of the Fe electronic structure in the $^1\pi-\pi^*$ state, we do not expect an appreciable difference signal. Consequently, the appearance of a time-dependent difference signal distinct from the $^5$MC state provides evidence for additional states involved in the ES relaxation dynamics. Shown in Fig. 2b–d is the difference signal during the first 850 fs after excitation. The spectral shape of the difference signal we observe for the early delay times (Fig. 2c) is clearly distinct from the difference signal at later delay times and strongly indicates the presence of a short-lived ES distinct from the $^1\pi-\pi^*$ and $^5$MC ES. As the reference spectra for different spin configurations shown in Fig. 2a demonstrate, the delayed appearance of a negative difference signal at 7054 eV compared to 7058 eV, shown in Fig. 2d, indicates that this intermediate state has either a doublet, triplet or quartet configuration.

Performing a detailed analysis of this observation by using the approaches described in Supplementary Note 1, we confirm the presence of a short-lived intermediate in the relaxation mechanism and assign it to a triplet metal-centered ($^3$MC) state. We then use the reference spectra shown in Fig. 2a to fit the time-resolved XES difference spectra using a rate equation model (Fig. 3). Our

Kβ XES measurement does not have a spectroscopic signature for the $^1\pi-\pi^*$ ES, so we use the exponential lifetime of 145 ± 5 fs measured by Bräm et al. for the $^1\pi-\pi^*$ ES[13]. We use the 5.9 ps exponential lifetime for the $^5$MC state measured by Mara et al.[8], leaving the lifetime of the intermediate state, the FWHM and time zero of the experimental response function, as well as the excitation yield as the free parameters in the analysis. The best fit gives an instrument response function FWHM of 118 ± 61 fs. The $^3$MC lifetime is fitted to 87 ± 51 fs and the excitation yield is fitted to 74 ± 2%. Despite the high excitation yield, we have been able to demonstrate the observed dynamics conform to those measured at lower excitation fluences, where a direct comparison can be made. Supplementary Note 2 has a detailed discussion of the power dependence. Fit results are summarized in Supplementary Tables 1–2.

A variety of potential intermediate electronic ES have been invoked in axial ligand photodissociation mechanisms for heme proteins. Both LMCT and MLCT states have been proposed to be involved. LMCT would produce low-spin Fe(I) and MLCT would produce low-spin Fe(III), both of which would be Fe spin doublets. Franzen et al. have proposed the first electronic ES transition for CO hemoglobin involves electron transfer from the occupied $d_\pi$ ($d_{xz}$, $d_{yz}$) orbital to the porphyrin $\pi$ orbital

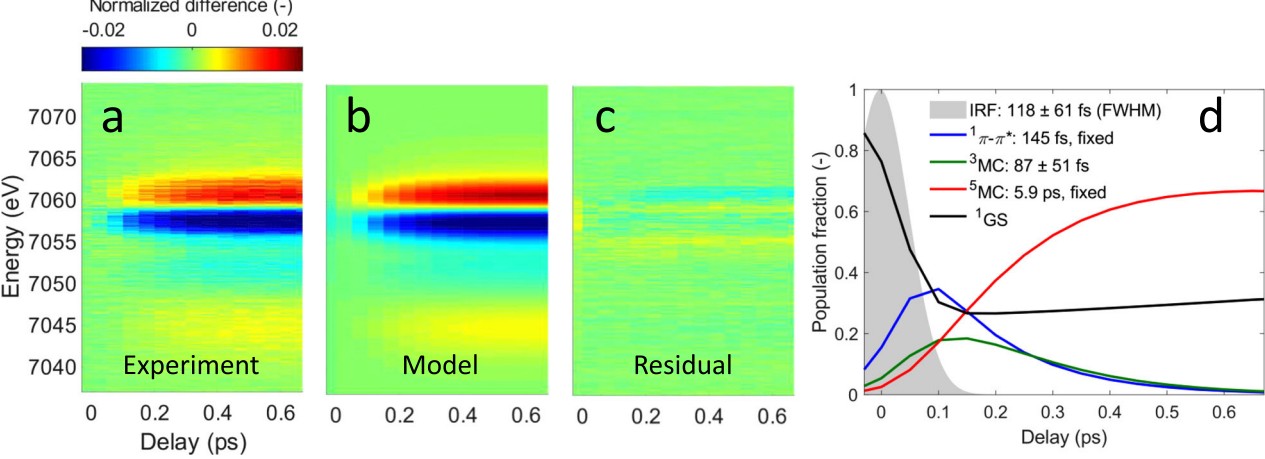

**Fig. 3 Population analysis from Fe Kβ X-ray emission spectroscopy. a** Experimental Kβ X-ray emission difference signal. **b** Best fit using a model with a triplet intermediate state. **c** Fit residual. **d** Resulting time-dependent populations for the $^1\pi-\pi^\star$, $^3$MC and $^5$MC states and the ground state ($^1$GS). The $^1\pi-\pi^\star$ and $^5$MC lifetimes were fixed to 145 fs and 5.9 ps, respectively. Fitted time constants are 118 ± 61 fs for the instrument response function FWHM and 87 ± 51 fs for the $^3$MC lifetime.

vacated in the optically generated electronic ES[11]. This ES should also be energetically accessible in cyt $c$, but should not initiate dissociation of the Fe–S bond. Bräm et al.[13] have proposed the $^1\pi-\pi^\star$ ES decays through an electron transfer from the $\pi^\star$ ES to the unoccupied $d_{z^2}$ orbital to create an Fe(I) ion. This orbital has antibonding $\sigma^\star$ character consistent with the Fe–S dissociation, but pulse radiolysis measurements show the porphyrin ring will be reduced, rather than the iron atom in ferrous cyt $c$[15], making $^1\pi-\pi^\star$ relaxation to a LMCT state energetically inaccessible.

Theoretical studies of the photodissociation mechanism in CO-bound myoglobin also provide useful insight into the potential mechanism for Fe(II)–S photodissociation in cyt $c$. In computational studies, Waleh and Loew propose that Fe(II)–CO dissociation involves excitation of an electron from the $d_\pi$ ($d_{xz}$, $d_{yz}$) orbital to the $d_{z^2}$ orbital[26]. This study does not make clear the mechanism for populating such a $d_\pi^3 d_{z^2}^1$ configuration but they conclude from their calculated ES energies that the transition can be directly assigned to a $^1[d_\pi^3 d_{z^2}^1]$ ($^1$MC) state and does not require intersystem crossing to a $^3[d_\pi^3 d_{z^2}^1]$ ($^3$MC) state[26]. Similar conclusions have been drawn in the theoretical study by Falahati et al. with the additional complication of significant configuration interaction between the $^1\pi-\pi^\star$ and the $^1[d_\pi^3 d_{z^2}^1]$ metal-centered ES[27].

Using these prior studies of heme protein photodissociation and the constraints imposed by the ultrafast Kβ XES measurement, we conclude that we have observed the involvement of a $^3[d_\pi^3 d_{z^2}^1]$ state in the Fe(II)–S bond dissociation. Since Kβ XES cannot distinguish between the $^3[d_\pi^3 d_{z^2}^1]$ and $^3[d_\pi^3 d_{x^2-y^2}^1]$ states, the choice of the $^3[d_\pi^3 d_{z^2}^1]$ state and its importance to bond dissociation will be made clearer in the following sections. The XES measurement leaves the $^3[d_\pi^3 d_{z^2}^1]$ formation mechanism unclear. Most likely, sequential MLCT and LMCT electron transfers, with one involving an intersystem crossing, would result in the formation of the $^3[d_\pi^3 d_{z^2}^1]$ ES. A Förster energy transfer mechanism dictating the lifetime of the $^1\pi-\pi^\star$ ES seems unlikely because the $^1[d_\pi^3 d_{z^2}^1]$ metal-centered ES has minimal oscillator strength due to symmetry selection rules.

**Ultrafast X-ray solution scattering**. Figure 4a shows the transient XSS signal $\Delta S$ for scattering vectors $Q$ in the range 0.2–3.3 Å$^{-1}$ and pump-probe time delays up to 600 fs and Fig. 4b shows $\Delta S$ for

fixed time delays extending to 15 ps. Our analysis focuses on the structural changes occurring during Fe(II)–S bond dissociation. Figure 4a clearly shows a prominent reduction in scattering between 0.4 and 1.1 Å$^{-1}$ induced by photoexcitation that develops a characteristic shape and maximum amplitude faster than the rise in the quintet state population (Fig. 4c, blue and red curves) and faster than the 700 fs time constant estimated for the appearance of global protein structural changes based on the 14 Å cyt $c$ radius of gyration[28] and strain wave propagation velocity of ~20 Å·ps$^{-1}$ measured by Levantino et al.[23]. As shown in Supplementary Note 3, this difference signal shows negligible time-dependent changes in shape and decays with a 5.2 ± 1.0 ps lifetime, similar to the 5.9 ps time constant for Fe(II)–S bond reformation extracted from the XES measurement[8]. These observations support the assignment of the difference signal in this $Q$-range primarily to local structural changes associated with Fe-axial coordination, which has informed our structural modeling of the XSS signal.

Starting with a ferrous cyt $c$ solution structure[29], we use a model for the ultrafast nuclear dynamics that only considers specific structural motions focused on changes in the axial ligand positions while neglecting other structural changes at the heme and global protein structural relaxation. Such a model reflects the antibonding nature of the $^3[d_\pi^3 d_{z^2}^1]$ ES with respect to the axial ligands and minimizes the number of structural parameters, thus respecting the limited information content of XSS difference scattering curves[30,31]. The structural analysis here is constrained to the first 300 fs during which the axial bonding changes significantly, and changes in structure occurring at larger length scales unaddressed by our model will be of lesser importance. Based on previous structural studies of cyt $c$[8,32], the simulated scattering signal is modeled as a linear combination of the protein signal arising from structural changes at the heme and the water heating signal[30]. The heme structural changes are parameterized using the positions of Met80 and His18 residues (Fig. 4d). A more detailed discussion of the model implementation and limitations can be found in Supplementary Note 4. Systematic modifications of these structural parameters clearly demonstrate that the negative difference signal seen between 0.4 and 1.1 Å$^{-1}$ requires significant elongation of both the Fe-Met80 and the Fe-His18 bond lengths. Figure 4b shows a comparison between fits of our model and the measured data at selected time delays. Molecular dynamics (MD) calculations of CO photolysis from myoglobin also show a reduction in scattering intensity in this $Q$-range

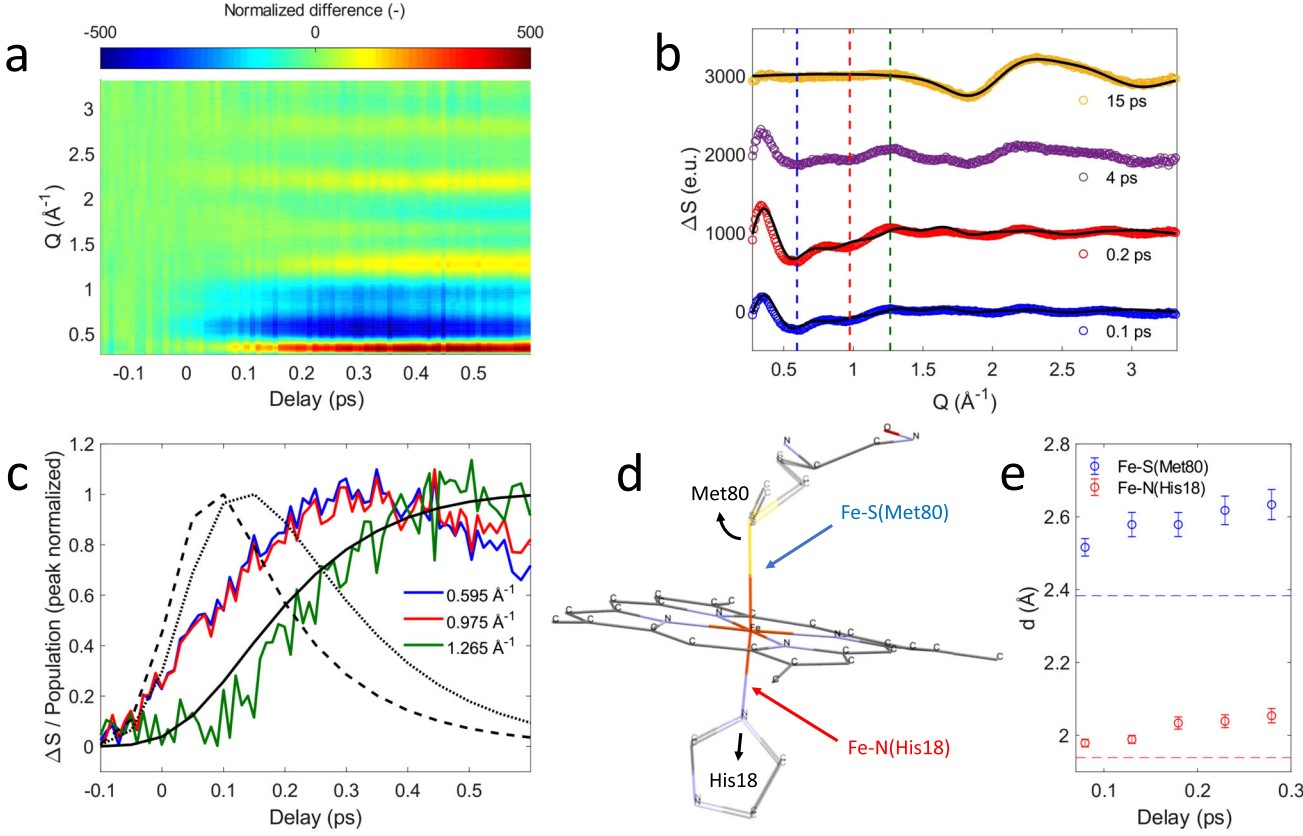

**Fig. 4 Modeling of the X-ray solution scattering difference signal. a** X-ray solution scattering (XSS) difference signal of ferrous cytochrome *c*. **b** XSS difference signal at different time delays. Black lines represent structural fits for the 0.1/0.2 ps curves as described in the text and the scaled bulk water heat differential for the 15 ps curve. **c** Time-dependence at *Q*-values indicated by dashed lines in **b**. Black lines represent the $^1\pi-\pi^\star$ (dashed), $^3$MC (dotted) and $^5$MC (solid) populations derived from the Kβ X-ray emission spectroscopy measurement (see Fig. 3). All curves are peak normalized for comparison. **d** Local structural changes are parameterized via Met80 rotation and His18 translation as illustrated by the black arrows. **e** Time evolution of the Fe-S (Met80) and Fe-N(His18) distances. The width of the time bins has been further increased by a factor of 5 with respect to the data shown in **a** and **c**. Errors are estimated for each original time bin assuming 15% uncertainty in sample concentration, 5% uncertainty in the excitation yield and a small discretization error from the fit procedure, then propagated to obtain the errors for the larger time bins. Horizontal dashed lines represent ground state values of the Fe-S (Met80) and Fe-N(His18) bond distances.

directly associated with Fe–CO bond dissociation, supporting our attribution of this reduction in scattering intensity to Fe-Met80 bond dissociation[31]. Within the constraints of our model, we fit a range of Fe-Met80 and Fe-His18 bond lengths (Fig. 4e). For the first 300 fs, our model qualitatively reproduces the observed XSS difference signal of cyt *c* without the need to invoke heme doming[19,33]. This is consistent with the observed delayed appearance of the $^5$MC state that has been suggested as the primary origin of the doming motion due to the antibonding nature of the singly occupied $d_{x^2-y^2}$ orbital with respect to the Fe (II)-N(Por) bonds[27,34,35]. We do observe a delayed rise in the positive peak at $Q = 1.265\ Å^{-1}$ strongly correlated with the rise time for quintet state formation (Fig. 4c, green and solid black curves), but attempts to capture this structural feature with heme doming have not been successful and indicate the $Q = 1.265\ Å^{-1}$ signal results from multiple structural changes. The fitted amplitudes of the Fe-Met80 and Fe-His18 structural parameters depend on how accurately the experimental data are rescaled to reflect a single liquid unit cell. A conservative estimate considering uncertainties in sample concentration (±15%) and excitation yield (±5%) dictates the error bars shown in Fig. 4e. When extending the analysis to 600 fs time delays, the Fe(II)-S distance can be reasonably fit with values between 2.5 Å and 2.7 Å, but not with the ≥3 Å found with the XANES analysis presented by Mara et al.[8]. Since a 300–600 fs doming motion

would further increase the Fe(II)-S distance, this difference (discussed in Supplementary Note 4) likely reflects the need for a more detailed model of the structural dynamics including the heme doming motion and related global structural changes after 300 fs[8]. These constraints on the analysis do not weaken the conclusion that the formation of a $^3$MC ES initiates the Fe–S bond dissociation.

The transient signal prevailing beyond 10 ps exhibits the well-known change in the bulk water structure factor resulting from ultrafast energy transfer and equilibration to an elevated solvent temperature[30,36]. The observed energy transfer to the solvent accesses the dynamics of energy transfer and equilibration between the protein and solvent. Mara et al.[8], supported by the rate of local equilibration found in MD simulations of photo-excited heme proteins[37], assumed local thermal equilibrium when analyzing the rate of six-coordinate singlet state reformation in ferrous cyt *c*, an assumption that has been questioned by Benabbas and Champion[38]. Here we use temperature-dependent changes in the water structure factor to investigate the time scale for local equilibration between protein and solvent. The structure of the difference scattering in the measured range does not show significant variation beyond 1 ps, though the amplitude of the difference signal does show the expected signal decay with a 5–6 ps time constant associated with the $^5$MC decay and a signal rise with a 7 ps time constant associated with the increasing water

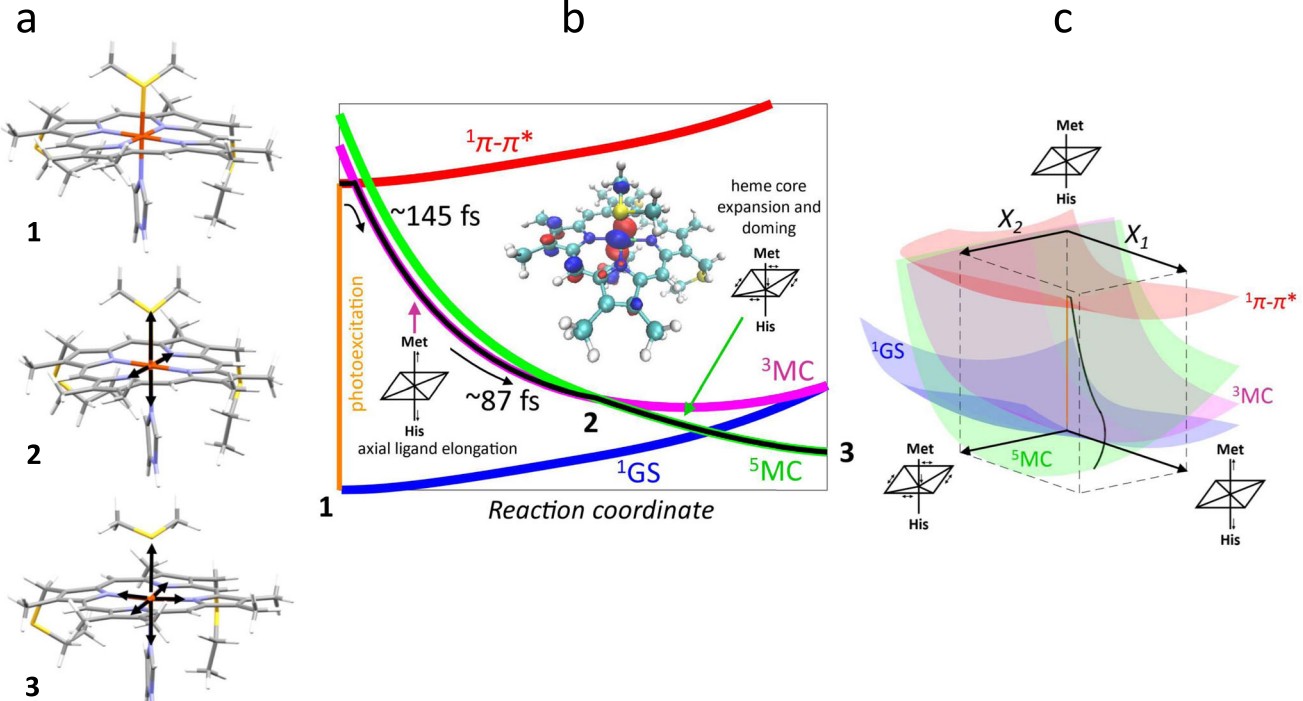

**Fig. 5 Schematic of the Fe–S bond dissociation and proposed kinetic model for the electronic states involved in bond photolysis of ferrous cytochrome c. a** Structures are shown of the optimized ground state (1, position indicated in **b**), triplet state at the crossing point (2, indicated in **b**), and the optimized quintet state (3). Black arrows indicate structural changes with respect to the GSS. **b** The vertical orange line represents the photoexcitation process and the black line is the proposed trajectory involving the $^1\pi$–$\pi^*$, $^3$MC, and $^5$MC states. The dominant motions on the $^3$MC and $^5$MC surfaces are indicated and defined as separate axes in **c**. The middle insert shows that the $d_\pi$ hole of the triplet in the GSS is aligned along an Fe-N(Por) axis, causing equatorial elongation leading to surface crossing. **c** 3D scheme of the proposed trajectory. The coordinate $X_1$ represents axial ligand elongation. $X_2$ comprises heme core expansion and doming. As in **b**, the vertical orange line represents the photoexcitation process and the black line is the proposed trajectory through the relevant ES.

temperature (see Supplementary Note 3). This ~7 ps time constant agrees with the ~7 ps time constant used for the local heme temperature by Mara et al.[8] and is consistent with the rate of energy equilibration found in the MD simulation by Zhang and Straub[37]. These observations support the conclusion that the structural degrees of freedom contributing to the structure factor in the measured Q-range equilibrate to the five-coordinate quintet state prior to relaxation back to the electronic ground state associated with Met80 rebinding and reformation of the Fe (II)–S bond.

**Correlation to calculations.** We performed Density functional theory (DFT) calculations to ascertain the electronic structures of the triplet and quintet states involved in ligand photodissociation. We optimized the singlet ground state structure (GSS), and generated triplet and quintet equivalents at this singlet GSS. The triplet state at the GSS contains an additional $d_{z^2}$ electron, promoted from a $d_\pi$ orbital. The quintet state at the GSS is ~5 kcal/mol higher than the triplet species at the GSS, as it now additionally has the $d_{x^2-y^2}$ orbital singly occupied. To investigate where these triplet and quintet surfaces cross, a series of energy calculations were performed at various elongated Fe-axial bond lengths (Supplementary Note 5, Supplementary Fig. 8). For all calculated Fe-axial bond lengths, the quintet surface remains ~4–6 kcal/mol above the triplet surface, indicating that an additional reaction coordinate is required for crossing between these two states. To determine the crossing point, geometry optimizations on the triplet and quintet structures were performed, beginning from the singlet ground state geometry

(Supplementary Note 5, Supplementary Fig. 9). These iterative geometry optimizations led to an energy crossing in very few steps, giving triplet and quintet species with similar geometric but different electronic structures, and eventually leading to optimized, low-lying triplet and quintet species with similar energies above the singlet ground state. It should be noted that the triplet state, both at higher energy in the ground geometry and in the relaxed triplet state, has an empty $d_{x^2-y^2}$ orbital, in contrast to the proposal by Benabbas and Champion[38]. The triplet and quintet structures at the energy crossing are shown in Fig. 5a. The triplet species exhibits asymmetric equatorial elongation due to the alignment of the $d_\pi$ hole along the equatorially-trans Fe-N(Por) ligands (Fig. 5b). The ramifications of this are discussed below.

## Discussion

Reliably detecting short-lived electronic excited states involved in ligand photolysis of heme compounds via femtosecond optical spectroscopy remains challenging[13,14,39–42]. Generally, MC ES have been invoked to reconcile the in-plane electronic redistribution associated with the $^1\pi-\pi^*$ photoexcitation with dissociative motions along the orthogonal axial ligand coordinate. Previous studies on CO photolysis from myoglobin have suggested that ultrafast CO dissociation may involve very short-lived, low-lying triplet ES[39,40]. However, these findings relied on earlier theoretical work[26] and symmetry and ligand-field considerations without direct support from experimental signatures for these states. Accordingly, photolysis of the Met80 ligand from ferrous heme in cyt c using either Soret or Q-band excitation has previously been proposed to occur from a dissociative metal-centered

ES heretofore unobserved experimentally. Using femtosecond resolution Kβ XES, we have identified a short-lived triplet metal-centered intermediate state with a $^3[d_\pi^3 d_{z^2}^1]$ configuration. Population of this state occurs from the Q-band $^1\pi-\pi^*$ ES and decays with an 87 fs lifetime to an Fe(II) quintet electronic ES with a $^5[d_\pi^2 d_{z^2}^1 d_{x^2-y^2}^1]$ configuration. The nuclear structural dynamics have been followed simultaneously with femtosecond XSS. Structural modeling assigns the reduction in scattering intensity between 0.4 and 1.1 Å$^{-1}$ to axial Fe(II)-S bond dissociation and axial Fe(II)–N bond elongation.

Based on these findings and the previous studies of Franzen et al.[11] and Falahati et al.[27], we propose the $^1\pi-\pi^*$ state generated by Q-band excitation decays via iron-to-porphyrin MLCT from the $d_\pi(d_{xz}, d_{yz})$ into the porphyrin π hole with a 145 fs time constant. MLCT state creation, and potentially intersystem crossing to the triplet ES manifold, enables prompt porphyrin-to-iron LMCT from the porphyrin $\pi^*$ to the predominantly Fe $d_{z^2}$ orbital generating the $^3$MC ES with a $^3[d_\pi^3 d_{z^2}^1]$ configuration. The $d_{z^2}$ orbital populated in this ES has $\sigma_{d_{z^2}}^*$ dissociative character with respect to the Fe(II)-S σ bond and initiates bond dissociation. In the Kβ XES experiment, this $^3$MC is the first electronic ES observed; we do not have spectroscopic evidence for the sequential or concerted MLCT and LMCT steps that we propose for the formation of the $^3$MC ES. Additionally, alignment of the dπ hole along the Fe-N axis results in loss of backbonding[43], causing equatorial expansion in the triplet state. Thus, $d_{z^2}$ occupation effectively causes both axial and equatorial elongation, enabling the triplet and quintet surfaces to cross with spin-orbit coupling and populating the quintet surface. This promotes a second $d_\pi$ electron into the $d_{x^2-y^2}$ orbital, causing additional heme core expansion and relaxation to the five-coordinate structure observed by Mara et al.[8]. This structure has both axial elongation and equatorial heme core expansion and doming, as observed experimentally with XAS and consistent with the slower component in the XSS signal (Fig. 4c). The proposed photolysis process $^1$GS (6C) $\xrightarrow{h\nu}$ $^1\pi-\pi^*$ (6C) $\xrightarrow{145\,fs}$ $^3$MC (dissociative) $\xrightarrow{87\,fs}$ $^5$MC (5C) $\xrightarrow{5-6\,ps}$ $^1$GS (6C) is illustrated in Fig. 5. These mechanistic studies of Fe(II)-Met80 dissociation set the stage for investigating how genetic modifications in the protein structure influence the dynamics of Fe(II)-Met80 rebinding and how the protein environment surrounding the heme influences protein function.

## Methods

**Experiment.** Solutions of horse heart cyt *c* were purchased from Sigma–Aldrich and prepared at 3–4 mM in 100 mM, pH 7.2 phosphate buffer. Cyt *c* was purified by FPLC on a cation exchange column using NaCl as elutant. Purified protein was dialyzed for ~24 h and then concentrated to 3–4 mM, as determined by UV-Vis spectroscopy. Solutions were reduced using sodium dithionite immediately before the experiment. Complete reduction was confirmed by changes in the UV-Vis spectrum (appearance of 520 nm and 550 nm bands, disappearance of 695 nm band). The Kβ XES and XSS data were collected during two different experimental runs at the X-ray Pump Probe (XPP) instrument[44] at the Linac Coherent Light Source (LCLS). Experimental details for the Kβ XES measurements were previously described in Mara et al.[8]. The sample was flowed through a 100 μm inner diameter capillary to form a ~100 μm diameter cylindrical liquid jet, using an HPLC pump. The sample was optically pumped and probed by 8 keV self-amplified stimulated emission (SASE) X-ray pulses (~$10^{12}$ photons/pulse, 120 Hz, 50 fs) shortly after exiting the capillary in the region of laminar flow. The jet was held under helium atmosphere, preventing oxidation to ferric cyt *c*. Optical excitation was performed nearly collinearly to the X-rays with 50 fs FWHM, 520 nm laser pulses (~20 mJ/cm$^2$) generated by optical parametric amplification of the 800 nm output of a Ti:sapphire regenerative amplifier laser system (Coherent, Legend). The pump laser fluence was determined by a power titration measurement at the beginning of the experiment and chosen to maximize the excited-state fraction while minimizing multiphoton absorption effects. The time delay between the laser and X-ray pulse was determined via the timing tool installed at XPP[44]. The X-ray pulses were focused using Be compound refractive lenses to a 50 μm diameter spot size on the sample jet. A high-energy resolution X-ray emission spectrometer, based on the von Hamos geometry, was used to capture the Fe Kβ XES signal[45]. The spectrometer was equipped with 4

cylindrically bent (0.5 m radius) Ge(620) crystal analyzers and set to cover the Bragg angle range from 78.1° to 80.5° corresponding to an energy range of 7.027 to 7.083 keV. A 140k Cornell-SLAC Pixel Array Detector[46] (CSPAD, 388 × 370 pixels) collected the Bragg diffracted X-rays. During the second experiment, both the Kβ XES and XSS data were measured simultaneously and the two resulting Kβ XES datasets were temporally aligned (Supplementary Note 6). A similar setup as during the first experiment was used for sample delivery with 3–4 mM solutions of cyt *c* flowing in a slightly smaller 75 μm diameter cylindrical liquid jet. Optical excitation was performed using the same wavelength and fluence. The Kβ XES data were collected using an ePix100 detector[46]. To detect the XSS data, a 2.3 M CSPAD[46] was used in forward scattering geometry. Full 2D images of the XES and XSS detectors were read out shot-to-shot and subsequently processed and binned according to their pump-probe delay. XES spectra were extracted by integrating the intensity in a rectangular area of interest containing a few pixels along the non-dispersive axis. The emission energy was calibrated by matching the laser off spectrum to a singlet reference spectrum[20].

**Theory.** Models for singlet, triplet, and quintet species were generated by DFT calculations. Ground state geometry optimizations were performed using Gaussian 09[47], with the unrestricted functional BP86, modified to include Hartree-Fock (HF) mixing of 20% with a triple-zeta (6–311 G*) basis set on Fe, N, and S, and a double-zeta (6–31 G*) basis set on all other atoms, as used in our previous studies[8,48]. This model includes the cross-linked cysteine side chains on the heme, which were kept fixed during geometry optimizations. This model was derived from cyt *c* crystal structure 1HRC[49]. Population analysis of the optimized structures was performed using Gaussview, and molecular orbital images were generated using VMD[50].

## Data availability
The XES and XSS data shown in Figs. 2b and 4a are provided as Source Data files. Source data are provided with this paper.

## Code availability
All relevant data and analysis scripts used in this study are available from the corresponding authors upon reasonable request.

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

## Acknowledgements

M.E.R., K.K. and K.J.G. acknowledge support from the U.S. Department of Energy, Office of Science, Basic Energy Sciences, Chemical Sciences, Geosciences, and Biosciences Division. This research was also supported by the National Institute of General Medical Sciences under awards R01GM040392 (E.I.S.) and F32GM122194 (L.B.G.). Use of the Linac Coherent Light Source (LCLS) and the Stanford Synchrotron Radiation Light-source (SSRL) of the SLAC National Accelerator Laboratory is supported by the U.S. Department of Energy (DOE) Office of Science, Office of Basic Energy Sciences under contract DE-AC02-76SF00515. The SSRL Structural Molecular Biology Program is supported by the DOE Office of Biological and Environmental Research and by the National Institutes of Health, National Institute of General Medical Sciences (P41GM103393). R.G.H. acknowledges a Gerhard Casper Stanford Graduate Fellowship and an Achievement Rewards for College Scientists Fellowship.

## Author contributions

K.J.G., E.I.S., R.G.H., M.W.M., U.B. and R.A.M. designed the experiments. M.E.R., M.W.M., T.K., H.L., R.G.H., R.A.M., T.B.v.D., M.C., J.M.G., S.N., D.S., K.K., R.W.H., C.W. and L.B.G. conducted the experiment at the LCLS. M.E.R. and K.J.G. analyzed the data with help from K.S.K. and E.B. M.W.M. performed DFT calculations and purified and prepared protein samples. K.J.G., M.E.R., M.W.M. and E.I.S. wrote the manuscript with input from all authors.

## Competing interests

The authors declare no competing interests.
