## [Peer Review File · Nature Communications]

Reviewer #1 (Remarks to the Author):

In this study, the authors report a study investigating the photodissociation dynamics of the Fe-S bond dissociation of a model protein in terms of both energies and structures using x-ray emission spectroscopy (XES) and x-ray solution scattering (XSS). The authors tracked the movement of the coordinated residues, Met18 and His80 using the XSS data, as well as the change of the spin states of the center iron using the XES data. Specifically, they identified the 3MC intermediate state which is formed before the penta-coordinated 5MC state is formed. Based on the data, they also propose that the coordinated Met18 and His80 residues move away from the Fe atom of the heme over time before 300 fs. The analysis for extracting the energetic information from the XES data was performed logically and the related discussion is also well described in text. In contrast, regarding the structural information obtained from XSS, this reviewer has some doubts and concerns about the structural analysis scheme that the authors used in this study. In this regard, the authors need to fully address the questions and concerns listed in the following.

The authors stated that “The 3MC lifetime is fitted to 87 ± 51 fs implying a quintet state formation time of ~ 180 fs ...”. Related to this statement, it is necessary to explain how the quintet state formation time (~ 180 fs) was calculated from the 3MC lifetime of 87 fs. The readers will be curious about this point since the kinetic analysis seems to have been conducted based on a sequential model including the transition from 3MC to 5MC. A more detailed explanation on how the authors established the overall kinetics from the XES data will be useful.

The authors mentioned that “The difference signal grows in within 500 fs, and as shown in the SI it decays with an exponential time constant of 5.2 ± 1.0 ps in the Q-range $0.3 - 1.3 \text{ \AA}^{-1}$, very similar to the 5.9 ps lifetime of the five-coordinate heme species measured with XES.¹¹ Consequently, we ascribe the early time scattering difference signal in this Q-range to structural changes of the heme and the protein residues coordinated to the Fe.”. Related to this statement, they assumed that the XSS signal change in several picoseconds was induced by the structural change around the heme and the structural analysis using the XSS data was also carried out using the model that involves changes only in the positions of the residues coordinated to the heme. In this regard, one cannot rule out the possibility that global structural changes of proteins can be accompanied even within a few picoseconds. Based on the previous optical study (Zhong et al., *JACS*, 2009, 2846), six-coordinated cyt c showed such impulsive bond breaking between heme and Met residue within 100 fs and also this local structural change generated global quake-like motion involving the dissipation of laser-induced thermal energy. In myoglobin, another heme protein, it has been reported that overall structural changes occur within 1 ps in a previous X-ray crystallography study (Schlichting et al., *Science*, 2015, 350, 445). In addition, in a previous XSS study (Cammarata et al., *Nat. Commun.*, 2015, 6772), it was observed that the energy absorbed by the heme in Mb spreads to the whole

protein and causes the global structural changes, showing an ultrafast quake-like motion. Considering these reports and the lack of direct evidence to support the localized structural motion without the global structural change, the current interpretation and structural analysis modeling presented by the authors regarding the XSS signal change in several picoseconds needs to be revised.

According to the previous publications (Kruglik et al., JACS, 2004, 13932 & Scopigno et al., JACS, 2020, 2285), six-coordinated cyt c showed the out-of-plane motion, so-called heme doming, on sub-picosecond regime during the photodissociation of Met80 residue. The authors described as follows in the SI; "... The Fe out-of-plane motion was parameterized by translating the Fe atom along the Fe(II)-N(His18) axis..... " It seems that the position of nitrogen atom was only changed along the specific axis without any rearrangements of neighboring pyrrole rings. Furthermore, the modelled structures displayed in Figures 4 and 5 have the planar heme core. In this regard, it is unclear how well the physically-reasonable heme doming motion was reflected in the modelled structures. To fully consider the previous results, it is necessary to consider the out-of-plane motion in the heme moiety by employing reasonable structures.

The authors should improve the readability of the SI. For example, there are two sets of Tables S1-S4.

Reviewer #2 (Remarks to the Author):

In this manuscript, the authors provide a detailed spectroscopic analysis of the progression of excited states that lead to breakage of the Fe-S(Met80) bond of ferrocycochrome c. In previous work, these authors have identified a quintet state that populates within about 500 fs following photoexcitation of the heme. In the current work, by combining XES and X-ray solution scattering they are able to connect electronic structure changes to related nuclear motions and clearly define an intermediate state that precedes the quintet metal-centered (MC) state. Using XES spectra of model compounds, they assign this state to a triplet MC state. Modeling the time-resolved data, they show that population of the intermediate state maximizes at about 100 fs followed by

conversion to the quintet MC state. The authors then use DFT calculations coupled to geometry optimization to demonstrate the feasibility of the progression of electronic states that lead to Fe(II)-S(Met80) bond scission following excitation. Furthermore, the authors note that, with this knowledge, the heme environment can now be tuned so that a photo-activatable switch that would turn cytochrome c into a peroxidase could be developed. This is a carefully done study that significantly advances our knowledge of how photoexcitation affects the electronic structure and bonding of the heme of cytochrome c. A few comments follow.

1. Looking at the residual plots in Fig. S3e it is not clear that the fit to a quartet state is much different than the fit to the doublet state. It would be useful to have a fuller analysis of how well the quartet fits the XES data added to Table S2.

2. In Table S2, the difference in the Residual Sums of Squares (RSS) are close enough that an extra decimal place in the RSS values would be useful.

3. Given that other assignments have been made for the MC excited state (Chergui and co-workers) that follows the singlet π^* state, it would be useful to have a quantitative assessment based on the residuals to the fits in Fig. S3 and Table S2 for the confidence of the assignment to the triplet MC versus doublet and quartet MC states. Do the data support the assignment with 90% confidence or is it 70%.

4. In Figure 4e, the y-axis legend should be changed to d, rather Δd , because an actual bond length is being reported in the figure, not the change in the length of the bond.

5. The authors should comment on whether the Fe-S bond length in the triplet MC state in Fig. 4e (~ 2.6 Å) corresponds to a broken bond or simply a weakened bond (based on known literature values). Or does the bond not really break until the quintet MC state forms (bond length increases to >3 Å based on XANES data)?

6. In the Discussion section, further elaboration of how the triple MC state forms would be useful. Both the current work and the work of Chergui and co-workers indicate that the electron in the π^* orbital migrates to the d_{z^2} orbital, which is at higher energy and therefore an unfavorable transition. The current work by identifying the intermediate state as a triplet state requires that another electron also move to form the triplet state. The triplet state can be accomplished by moving a $d\pi$ electron into the π hole. The $d\pi$ to π transition would provide the needed thermal energy to promote the π^* electron to the d_{z^2} orbital. A more detailed outline of this process would be useful, particularly for the more general audience of Nature Communications.

7. In the Discussion, the authors talk about the potential to use mutagenesis to manipulate the heme environment to create a photoinducible switch that would convert cytochrome c into a peroxidase. The other piece to this challenge is the Met80 rebinding rate. Some discussion of how fast Met80 rebinds to the heme and how much it would need to be slowed to generate an effective switch might be useful in this context, too.

Reviewer #3 (Remarks to the Author):

NCOMMS-20-29717-T

Short-lived metal-centered excited state initiates iron-methionine photodissociation in ferrous cytochrome c

The present study aimed at identifying the electronic excited states of the heme iron involved in the dissociation of the internal axial ligand Met80 (amino-acid side chain) in mitochondrial cytochrome c. This identification was performed by means of ultrafast (50 fs pulses) x-ray solution scattering and x-ray emission spectroscopy. The authors identified a triplet state as dissociative, which appears in 145 fs and decays in 87 fs to the 5-coordinate dissociated iron (quintet state).

This subject has been addressed a long time ago in the case of the heme proteins myoglobin and hemoglobin bound with O₂ and CO (PNAS 1980, 77, 5606; Biochemistry 1988, 27, 4049) but not fully resolved. This study brings new information in the case of Cyt c by means of a yet scarcely used methodology. The data are convincing and their analysis is rigorous. I recommend the publication of this manuscript provided minor changes are performed, as described below.

At the end of first paragraph of Introduction, the authors wrote "Controlling the transformation of cyt c to a peroxidase enzyme with light could lead to photodynamic therapy applications...". I do not think that it is necessary to mention such highly hypothetical application. The present study is sufficiently interesting in itself and does not need such "advertisement".

Page 7.

Figure 3. In the right panel, one understands that the rise of the signal between 0 and 0.1 ps is due to the instrument function. Please indicate the IRF in the legend or, better, in the panel with a graphic mean (maybe the calculated contour of the IRF). The fitted lifetimes can be indicated in the panel.

Can the FWHM and IRF be determined by an experimental mean rather than by fitting? For example by using a sample which has a simple "instantaneous" response with respect to the IRF. This would also determine the time zero position.

Since both pulses have a duration of 50 fs, one could expect a shorter IRF. Are there hardware elements (electronics or optics) which influence the IRF? Such information can be useful.

Page 8, last paragraph.

The authors must clearly indicate that the 5.9 ps time constant corresponds to Met80 rebinding and Fe – S bond reformation.

Figure 4: Is the IRF the same for XES and XSS ?

Page 9, line 23.

The authors wrote "Including the iron out-of-plane doming as a structural parameter does not improve the fit". Yes, it is not unexpected. This is a strong indication that heme doming occurs simultaneously with the measured electronic transition. Indeed, it is known that after dissociation of the distal axial ligand from 6-coordinate hemes, the change of the Fe spin drives immediately a change of the porphyrin core size, then an out-of-plane motion of the Fe and of the proximal His.

Page 11, line 11 (end of first paragraph).

It must be made clear that the relaxation of the quintet state to the ground state corresponds to Met80 rebinding to the heme Fe.

Discussion

The discussion is agreeably concise. Overall, I agree with the interpretations made by the authors. I appreciate the figure 5 which is rich in information and clearly summarizes the results, both experimental and theoretical. In the central panel the black line, which is important, is not easily seen. Maybe there is a way to better show the trajectory.

I would like to suggest to add somewhere in the discussion a scheme like this one:

+hv 145 fs 87 fs ~6 ps

1GS → 1π-π* → 3MC → 5MC → 1GS

6C 6C dissociative 5C 6C (Fe coordination state)

in order to extend the readership to readers who are not familiar with potential energy surfaces.

Since the 3MC excited state has a dissociative electronic configuration, the authors reasonably inferred that the rupture of the Fe-S bond occurs in this state. They should further discuss this result with respect to previous studies using femtosecond time-resolved visible absorption spectroscopy, which is sensitive to the electronic state of the heme (JPCB 2006, 110, 12766; JPCA 2003; 107, 8156). The photophysics and the nature of the electronic states in heme-ligand photodissociation has been investigated for decades. The authors should discuss seminal works which identified triplet charge transfer states involved in photodissociation (PNAS 1980, 77, 5606; Biochemistry 1988, 27, 4049).

Page 13: Again, photo-triggering apoptosis is a highly hypothetical application, since the lifetime of photodissociated Met80 is ~5-6 ps, the population of "apoptosis-active" Cyt c would be very low. Phototherapy is not the subject of this study and this sentence should be removed. (For apoptosis, analogs of cardiolipin as Cyt c ligands could be a better possibility).

The Supplementary Informations are complete and useful.

REVIEWER COMMENTS

Reviewer #1 (Remarks to the Author):

In this study, the authors report a study investigating the photodissociation dynamics of the Fe-S bond dissociation of a model protein in terms of both energies and structures using x-ray emission spectroscopy (XES) and x-ray solution scattering (XSS). The authors tracked the movement of the coordinated residues, Met18 and His80 using the XSS data, as well as the change of the spin states of the center iron using the XES data. Specifically, they identified the 3MC intermediate state which is formed before the penta-coordinated 5MC state is formed. Based on the data, they also propose that the coordinated Met18 and His80 residues move away from the Fe atom of the heme over time before 300 fs. The analysis for extracting the energetic information from the XES data was performed logically and the related discussion is also well described in text. In contrast, regarding the structural information obtained from XSS, this reviewer has some doubts and concerns about the structural analysis scheme that the authors used in this study. In this regard, the authors need to fully address the questions and concerns listed in the following.

The authors stated that “The 3MC lifetime is fitted to 87 ± 51 fs implying a quintet state formation time of ~ 180 fs ...”. Related to this statement, it is necessary to explain how the quintet state formation time (~ 180 fs) was calculated from the 3MC lifetime of 87 fs. The readers will be curious about this point since the kinetic analysis seems to have been conducted based on a sequential model including the transition from 3MC to 5MC. A more detailed explanation on how the authors established the overall kinetics from the XES data will be useful.

We appreciate the reviewer highlighting that our description of the kinetic model in the manuscript was unclear. To address this weakness, we have removed the comment about the quintet rise time and added a scheme for the kinetic model used in the analysis at the end of the discussion section (bottom of page 13).

The authors mentioned that “The difference signal grows in within 500 fs, and as shown in the SI it decays with an exponential time constant of 5.2 ± 1.0 ps in the Q-range $0.3 - 1.3 \text{ \AA}^{-1}$, very similar to the 5.9 ps lifetime of the five-coordinate heme species measured with XES.¹¹ Consequently, we ascribe the early time scattering difference signal in this Q-range to structural changes of the heme and the protein residues coordinated to the Fe.”. Related to this statement, they assumed that the XSS signal change in several picoseconds was induced by the structural change around the heme and the structural analysis using the XSS data was also carried out using the model that involves changes only in the positions of the residues coordinated to the heme. In this regard, one cannot rule out the possibility that global structural changes of proteins can be accompanied even within a few picoseconds. Based on the previous optical study (Zhong et al., JACS, 2009, 2846), six-coordinated cyt c showed such impulsive bond breaking between heme and Met residue within 100 fs and also this local structural change generated global quake-like motion involving the dissipation of laser-induced thermal energy. In myoglobin, another heme protein, it has been reported that overall structural changes occur within 1 ps in a previous X-ray crystallography study (Schlichting et al., Science, 2015, 350, 445). In addition, in a previous XSS study (Cammarata et al., Nat. Commun., 2015, 6772), it was observed that the energy absorbed by the heme in Mb spreads to the whole protein and causes the global structural changes, showing an ultrafast quake-like motion. Considering these reports and the lack of direct evidence to support the localized structural motion without the global structural change,

the current interpretation and structural analysis modeling presented by the authors regarding the XSS signal change in several picoseconds needs to be revised.

We thank the reviewer for raising concerns about potential inconsistencies between our description of the dynamics extracted from the x-ray scattering measurement and the mechanistic picture extracted from prior ultrafast optical and x-ray scattering measurements. The apparent inconsistencies result from lack of clarity in our discussion of the structural model we have used and our focus on not over interpreting the x-ray scattering measurement.

To address these concerns, we will present a detailed comparison between our work and those previous studies to demonstrate that our work adds to the previous studies and does not contradict prior measurements. The key questions regarding heme doming and change in the secondary structure of the protein will be discussed independently.

First, we will address the absence of the global structural response in our analysis of the XSS data. While we did not address them in the submitted manuscript, our study does not dispute the occurrence of global structural changes following methionine photolysis in cyt *c*. The reason we did not discuss these dynamics reflect the limited time (0-300 fs) and Q (0.3-4.5 \AA^{-1}) range of the analysis. Specifically, the experimental results and modeling indicate that the x-ray solution scattering difference signal in the 0 – 300 fs and 0.3 – 4.5 \AA^{-1} range exhibits limited sensitivity to these global structural changes and motivated our decision to exclude their discussion from the manuscript. Our approach is supported by the following observations:

- Zhong *et al.* have reported optical transient absorption signatures of cyt *c* in the UV range where the tryptophan residue absorbs. These transient signatures persisted for ~42 ps and were assigned to global protein conformation relaxation. In contrast, the dominant XSS difference signal occurring in the $Q = 0.4 - 1.1 \text{\AA}^{-1}$ range decays with a similar time constant of 5.2 ps as the five-coordinate quintet heme of 5.9 ps determined from the XES measurement. The similarity of these timescales combined with the unambiguous assignment of the latter 5.9 ps time constant to heme-methionine recombination support the assignment of the difference signal in the 0-300 fs and $Q = 0.4 - 1.1 \text{\AA}^{-1}$ range of our structural analysis to structural motions centered around the heme.
- Strain released upon photolysis propagates at ~20 \AA ps^{-1} in myoglobin (Cammarata *et al.*, Nature Comm., 2015, 6, 6772) and the radius of gyration for cyt *c* is $R_g \sim 12-14 \text{\AA}$ (Chapman *et al.*, Biochimica et Biophysica Acta – Biomembranes, 1969, 173, 1, 1). Based on these values we would expect global protein changes to grow in on a timescale of ~600 – 700 fs. In contrast, the characteristic shape of the XSS difference signal that we observe in the $Q = 0.4 - 1.1 \text{\AA}^{-1}$ range clearly grows in on a faster timescale, preceding the quintet formation timescale (Fig. 4d). The shape of the difference signal in the 0.4 – 1.1 \AA^{-1} range changes only slightly between the earliest and latest delays. These observations further support the assignment of these features to local structural changes. As outlined in the manuscript, it has previously been suggested (Huix-Rotllant *et al.*, Nature Comm., 2018, 9, 4502, Chergui *et al.*, PNAS, 2020, 117, 36, 21914, Martin *et al.*, Biochemistry, 1995, 34, 1224) that the quintet state is strongly coupled to the iron-out-of-plane motion and the latter in turn triggers global structural changes. This is consistent with the presence of features in the XSS difference signal (1.265 \AA^{-1} kinetic trace in Fig. 4d) that evolve on a similar timescale as the quintet population. Again, this supports the assignment of the XSS difference signal in the $Q = 0.4 - 1.1 \text{\AA}^{-1}$ range to local structural motions, occurring before the quintet state is significantly populated.

- The study of Cammarata *et al.* on photoexcited myoglobin had assigned transient signals in the small angle x-ray scattering region ($Q < 0.15 \text{ \AA}^{-1}$) to quake-like motions triggered by CO-photolysis, while their $0.4 - 1.1 \text{ \AA}^{-1}$ XSS difference signal was qualitatively associated with global structural motions in the context of pre-existing literature. These results are therefore not inconsistent with our interpretation of the XSS difference signal of photoexcited cyt c that does not include the $Q < 0.15 \text{ \AA}^{-1}$ range. In addition, the theoretical work of Brinkmann and Hub about CO-photolysis in myoglobin (Brinkmann and Hub, PNAS 2016, 113, 38, 10565) strongly supports our assignment of local structural motions as the primary origin of the difference signal in that Q-range. For myoglobin, they conclude that the negative feature around $Q = 0.75 \text{ \AA}^{-1}$ present in the XSS difference signal at the earliest delay times can be reproduced by CO-removal without considering the global protein structural response. They propose a signature of the protein quake located around 0.28 \AA^{-1} . Since the experiment of Cammarata *et al.* was conducted with a time resolution of ~ 500 fs, a separation of localized from global structural changes in their difference signal was not feasible.
- The work of Schlichting *et al.* (Schlichting et al., Science 2015, 350, 6259, 445) reports myoglobin shape changes occurring within 500 fs upon CO photodissociation. This timescale is comparable to the timescale on which the quintet population grows in as derived from our x-ray emission difference signal. Moreover, they observe biexponential heme doming with a sub-100 fs and a slower 400 fs component with $\sim 90\%$ of the doming occurring within 700 fs. These timescales and their interpretation are entirely consistent with the slower and more subtle components in our XSS difference signal. In our data, these slower transient changes are most pronounced around 1.27 \AA^{-1} (Fig. 4d) but do not distort the shape of the sub-300 fs difference signal in the range $Q = 0.4 - 1.1 \text{ \AA}^{-1}$ which we have assigned to localized structural motions.
- Based on all these considerations and the avoidance of overfitting the ‘information-poor’ XSS difference signal (Haldrup et al., Acta Crystallographica Section A, 2010, A66, 261-269, Brinkmann and Hub, PNAS, 2016, 113, 38, 10565), we have implemented a limited model guided by prior experimental studies and the clear correlation between the XSS difference signal and axial ligand-Fe bond elongation and Fe-S bond rupture. Our analysis then demonstrates that such a model is fully capable to account for the features observed in the XSS difference signal at delays prior to significant population of the quintet state. Attempts to include heme doming to explain the delayed appearance of the difference signal centered at 1.27 \AA^{-1} (Fig. 4d) did not succeed, perhaps indicating this signal originates, at least in part, from secondary structural changes. Our model does not include such effects, so we have focused on time delays that precede larger scale structural changes but capture the Fe-S bond rupture and enable us to correlate these structural changes with electronic excited state dynamics.

Following these bullet points we have significantly adjusted the presentation of the analysis in the manuscript.

According to the previous publications (Kruglik et al., JACS, 2004, 13932 & Scopigno et al., JACS, 2020, 2285), six-coordinated cyt c showed the out-of-plane motion, so-called heme doming, on sub-picosecond regime during the photodissociation of Met80 residue. The authors described as follows in the SI; “... The Fe out-of-plane motion was parameterized by translating the Fe atom along the Fe(II)-N(His18) axis. ...” It seems that the position of nitrogen atom was only changed along the specific axis without any rearrangements of neighboring pyrrole rings. Furthermore, the modelled structures displayed in Figures 4 and 5 have the planar heme core. In this regard, it is unclear how well the physically-reasonable heme doming motion was reflected in the modelled structures. To fully consider

the previous results, it is necessary to consider the out-of-plane motion in the heme moiety by employing reasonable structures.

Our efforts to quantify the doming coordinate for cytochrome *c* proved inconclusive. While we do not dispute the importance of doming in the full structural dynamics, we believe that our choice of the structural parameters at the earliest delays is physically meaningful as the observed ^3MC excited state has antibonding character predominantly with respect to the axial ligands. In contrast, the ^5MC state additionally has an electron in the $d_{x^2-y^2}$ type orbital and therefore has antibonding character for both, axial and equatorial bonds of the Fe, which has been proposed as the primary driving force for the doming motion (Franzen et al. *The Journal of Biological Chemistry*, 1995, 270, 4, 1718).

Since the characteristic shape of the XSS difference signal in the range $Q = 0.4 - 1.1 \text{ \AA}^{-1}$ grows in faster than the quintet population, such a driving force is lacking at the earliest delays. Our neglect of the doming motion during the first 300 fs is also consistent with the study of Huix-Rotllant et al. on CO-photolysis from myoglobin (Huix-Rotllant et al., *Nature Comm.*, 2018, 9, 4502), where they concluded that the doming motion stabilizing the high-spin state has a slow period of 371 fs and does not significantly affect the short-time dynamics of photolysis. We would also like to emphasize that this issue does not impact the dominant findings of the manuscript; specifically that the Fe-S bond elongation and dissociation is triggered by the formation of a ^3MC electronic excited state.

The authors should improve the readability of the SI. For example, there are two sets of Tables S1-S4. The labeling in the SI has been corrected accordingly.

Reviewer #2 (Remarks to the Author):

In this manuscript, the authors provide a detailed spectroscopic analysis of the progression of excited states that lead to breakage of the Fe-S(Met80) bond of ferrocycytochrome *c*. In previous work, these authors have identified a quintet state that populates within about 500 fs following photoexcitation of the heme. In the current work, by combining XES and X-ray solution scattering they are able to connect electronic structure changes to related nuclear motions and clearly define an intermediate state that precedes the quintet metal-centered (MC) state. Using XES spectra of model compounds, they assign this state to a triplet MC state. Modeling the time-resolved data, they show that population of the intermediate state maximizes at about 100 fs followed by conversion to the quintet MC state. The authors then use DFT calculations coupled to geometry optimization to demonstrate the feasibility of the progression of electronic states that lead to Fe(II)-S(Met80) bond scission following excitation. Furthermore, the authors note that, with this knowledge, the heme environment can now be tuned so that a photo-activatable switch that would turn cytochrome *c* into a peroxidase could be developed. This is a carefully done study that significantly advances our knowledge of how photoexcitation affects the electronic structure and bonding of the heme of cytochrome *c*. A few comments follow.

1. Looking at the residual plots in Fig. S3e it is not clear that the fit to a quartet state is much different than the fit to the doublet state. It would be useful to have a fuller analysis of how well the quartet fits the XES data added to Table S2.

We have extended the analysis shown in the supplementary information in Tab. S2 by adding Akaike information criterion differences and Akaike weights. Established guidelines for these values allow quantifying how well a specific model performs relative to the best model in the considered set of

models. Using the Akaike weights, we conclude that the triplet model is ~1.6 times more likely to be the best model than either the doublet or the quartet model.

2. In Table S2, the difference in the Residual Sums of Squares (RSS) are close enough that an extra decimal place in the RSS values would be useful.

We have added an extra decimal place in the RSS values of Tab. S2.

3. Given that other assignments have been made for the MC excited state (Chergui and co-workers) that follows the singlet π^* state, it would be useful to have a quantitative assessment based on the residuals to the fits in Fig. S3 and Table S2 for the confidence of the assignment to the triplet MC versus doublet and quartet MC states. Do the data support the assignment with 90% confidence or is it 70%.

We have added a more quantitative comparison of the doublet, triplet and quartet models to Tab. S2 by evaluating the Akaike information criterion differences and Akaike weights for each model. This analysis allows for comparing non-nested models and provides an empirical guideline for model selection. Based on this analysis, our assignment of the triplet intermediate is further supported while we cannot discard alternative doublet or quartet intermediate models with certainty. Given the limitations of the assumptions that underlie our analysis of the time-resolved XES spectra, we do not believe this measurement independent of additional knowledge of the electronic structure of ferrous cytochrome *c*. With respect to the ligand photolysis mechanism proposed by Chergui and co-workers, we consider their proposed deactivation pathway of the $^1\pi-\pi^*$ excited state via a porphyrin-to-metal charge transfer forming a dissociative doublet or quartet Fe(I) intermediate as unlikely due to the energetic reasons discussed in our manuscript. Specifically, we do not think the energetics support a single electron transfer from the $\pi-\pi^*$ excited state to an Fe(I) electronic configuration. Alternatively, a metal-to-porphyrin charge transfer filling the porphyrin π -hole and transiently producing a doublet Fe(III) species cannot be excluded based on energetic arguments, but transferring the electron from the Fe $d\pi$ orbitals should provide little driving force for ligand dissociation because the Fe-S bond lacks π -character. However, such a metal-to-porphyrin charge transfer increases the effective charge on the iron and could therefore lower the Fe d_{z^2} orbital energy, making the porphyrin-to-metal charge transfer energetically feasible, thus enabling the formation of a dissociative singlet or triplet metal-centered excited state in an additional step. For these reasons, we support the conclusion that two electron transfer steps, first metal-to-ligand and then ligand-to-metal, lead to Fe-S bond dissociations. While this relaxation mechanism

is consistent with our measurement and prior studies, the experiment under determines all the parameters associated with this mechanism and motivated our use of the simplest, statistically supported model including a single short-lived 3MC intermediate state.

4. In Figure 4e, the y-axis legend should be changed to d , rather Δd , because an actual bond length is being reported in the figure, not the change in the length of the bond.

We thank the reviewer for pointing this out and have corrected it accordingly.

5. The authors should comment on whether the Fe-S bond length in the triplet MC state in Fig. 4e (~2.6 Å) corresponds to a broken bond or simply a weakened bond (based on known literature values). Or does the bond not really break until the quintet MC state forms (bond length increases to >3 Å based on XANES data)?

Using density functional theory, we have calculated the triplet and quintet potential energy surfaces along this coordinate (Fig. S9). These surfaces exhibit a dissociative profile according to the criterion of Waleh and Loew (Waleh and Loew, JACS 1982, 104, 2346) and one may therefore consider that the bond breaks as the system moves along the iron-ligand coordinate of the triplet surface. While attractive forces for the methionine ligand are absent once the system is on the triplet surface, the attachment of the methionine ligand to the protein backbone may impose steric constraints that can significantly limit the range of methionine motion away from the iron center. When the doming motion occurs, predominantly as a consequence of quintet state formation (the Fe $d_{x^2-y^2}$ type molecular orbital is antibonding with respect to the Fe(II)-N(Por) bonds resulting in porphyrin core expansion and doming), the Fe-S bond increases towards 3 Å, even if the methionine residue does not move further.

6. In the Discussion section, further elaboration of how the triple MC state forms would be useful. Both the current work and the work of Chergui and co-workers indicate that the electron in the π^* orbital migrates to the d_{z^2} orbital, which is at higher energy and therefore an unfavorable transition. The current work by identifying the intermediate state as a triplet state requires that another electron also move to form the triplet state. The triplet state can be accomplished by moving a $d\pi$ electron into the π hole. The $d\pi$ to π transition would provide the needed thermal energy to promote the π^* electron to the d_{z^2} orbital. A more detailed outline of this process would be useful, particularly for the more general audience of Nature Communications.

We agree with the reviewer that this is an intriguing question. However, since we are unable to identify any additional short-lived intermediate states from our XES data, we can only speculate about the various possible scenarios that could facilitate the excitation transfer from the porphyrin to the metal center. The invoked scenario may strongly depend on symmetry considerations which determine to which extent selection rules will apply, electronic-nuclear coupling that can modify relative energetics, and the level of theory used to describe the electronic excited states. The brief description in the XES results section of the manuscript was written with these limitations in mind.

7. In the Discussion, the authors talk about the potential to use mutagenesis to manipulate the heme environment to create a photoinducible switch that would convert cytochrome c into a peroxidase. The other piece to this challenge is the Met80 rebinding rate. Some discussion of how fast Met80 rebinds to the heme and how much it would need to be slowed to generate an effective switch might be useful in this context, too.

We have removed this statement from the discussion.

Reviewer #3 (Remarks to the Author):

NCOMMS-20-29717-T

Short-lived metal-centered excited state initiates iron-methionine photodissociation in ferrous cytochrome c

The present study aimed at identifying the electronic excited states of the heme iron involved in the dissociation of the internal axial ligand Met80 (amino-acid side chain) in mitochondrial cytochrome c. This identification was performed by means of ultrafast (50 fs pulses) x-ray solution scattering and x-ray

emission spectroscopy. The authors identified a triplet state as dissociative, which appears in 145 fs and decays in 87 fs to the 5-coordinate dissociated iron (quintet state).

This subject has been addressed a long time ago in the case of the heme proteins myoglobin and hemoglobin bound with O₂ and CO (PNAS 1980, 77, 5606; Biochemistry 1988, 27, 4049) but not fully resolved. This study brings new information in the case of Cyt c by means of a yet scarcely used methodology. The data are convincing and their analysis is rigorous. I recommend the publication of this manuscript provided minor changes are performed, as described below.

At the end of first paragraph of Introduction, the authors wrote "Controlling the transformation of cyt c to a peroxidase enzyme with light could lead to photodynamic therapy applications...". I do not think that it is necessary to mention such highly hypothetical application. The present study is sufficiently interesting in itself and does not need such "advertisement".

We have removed this statement.

Page 7.

Figure 3. In the right panel, one understands that the rise of the signal between 0 and 0.1 ps is due to the instrument function. Please indicate the IRF in the legend or, better, in the panel with a graphic mean (maybe the calculated contour of the IRF). The fitted lifetimes can be indicated in the panel.

We have updated the right panel of Figure 3 by including a visualization of the IRF. We have also extended the legend to include the fixed and fitted time constants discussed in the main text.

Can the FWHM and IRF be determined by an experimental mean rather than by fitting? For example by using a sample which has a simple "instantaneous" response with respect to the IRF. This would also determine the time zero position.

Yes, in principle we could have used a reference sample with a known 'instantaneous' response such as it is for instance the case for direct photoexcitation into a metal-to-ligand or ligand-to-metal charge transfer excited state measured under the same conditions during the same experiment. Since this has not been done for the data included in the manuscript, we rely on the excellent agreement between the IRF determined from fitting the experimental data (118 ± 61 fs) and the theoretically estimated IRF (~110 – 130 fs). The IRF estimate considers 50 fs optical laser and x-ray pulses and a group velocity mismatch between them on the order of 1.1 fs/ μ m as estimated using an index of refraction of 1.33 for water. For a 75-100 μ m thick sample jet, the estimated IRF is therefore

$$f(50 \text{ fs})^2 + (50 \text{ fs})^2 + (97 \text{ fs})^2 \approx 120 \text{ fs.}$$

Since both pulses have a duration of 50 fs, one could expect a shorter IRF. Are there hardware elements (electronics or optics) which influence the IRF? Such information can be useful.

We have achieved excellent agreement between the fitted IRF value based on the experimental data and the calculated IRF value. As outlined above, the estimated IRF relies on the optical laser and x-ray pulse durations and considers the group velocity mismatch resulting from propagation through the sample jet. We have corrected our initial estimate for the IRF in the SI (which was based on an incorrect estimate of the group velocity mismatch).

Page 8, last paragraph.

The authors must clearly indicate that the 5.9 ps time constant corresponds to Met80 rebinding and Fe – S bond reformation.

We have added this clarification.

Figure 4: Is the IRF the same for XES and XSS ?

Yes. As described above, the IRF depends on the optical laser and x-ray pulse durations and the group velocity mismatch via the sample jet thickness. Our experimental setup allows measuring XES and XSS simultaneously, therefore IRF and time zero are both identical for these datasets which greatly simplifies data treatment.

Page 9, line 23.

The authors wrote "Including the iron out-of-plane doming as a structural parameter does not improve the fit". Yes, it is not unexpected. This is a strong indication that heme doming occurs simultaneously with the measured electronic transition. Indeed, it is known that after dissociation of the distal axial ligand from 6-coordinate hemes, the change of the Fe spin drives immediately a change of the porphyrin core size, then an out-of-plane motion of the Fe and of the proximal His. We have clarified the discussion regarding the quintet population and the doming coordinate by emphasizing that the dominant part of the doming motion should result from population of the quintet state while our sub-300 fs XSS difference signal reflects a structure that predominantly experiences a driving force for axial ligand elongation but no significant driving force for the doming coordinate.

Page 11, line 11 (end of first paragraph).

It must be made clear that the relaxation of the quintet state to the ground state corresponds to Met80 rebinding to the heme Fe.

We have added this clarification in the manuscript.

Discussion

The discussion is agreeably concise. Overall, I agree with the interpretations made by the authors. I appreciate the figure 5 which is rich in information and clearly summarizes the results, both experimental and theoretical. In the central panel the black line, which is important, is not easily seen. Maybe there is a way to better show the trajectory. I would like to suggest to add somewhere in the discussion a scheme like this one:

+hv 145 fs 87 fs ~6 ps
1GS → 1π-π* → 3MC → 5MC → 1GS
6C 6C dissociative 5C 6C (Fe coordination state)

in order to extend the readership to readers who are not familiar with potential energy surfaces. We have added the proposed scheme at the end of the discussion section and thickened the lines in the central panel of figure 5 for better visibility.

Since the 3MC excited state has a dissociative electronic configuration, the authors reasonably inferred that the rupture of the Fe-S bond occurs in this state. They should further discuss this result with respect to previous studies using femtosecond time-resolved visible absorption spectroscopy, which is sensitive to the electronic state of the heme (JPCB 2006, 110, 12766; JPCA 2003; 107, 8156). The photophysics and the nature of the electronic states in heme-ligand photodissociation has been investigated for

decades. The authors should discuss seminal works which identified triplet charge transfer states involved in photodissociation (PNAS 1980, 77, 5606; Biochemistry 1988, 27, 4049).

We thank the reviewer for suggesting a more thorough discussion of previous femtosecond optical spectroscopy work. This discussion indeed highlights the difficulty to unambiguously detect and characterize such short-lived excited states in heme photophysics and chemistry and thus emphasizes the strength of Fe K β x-ray emission spectroscopy. We have added a paragraph dedicated to these prior studies directly at the beginning of the discussion section.

Page 13: Again, photo-triggering apoptosis is a highly hypothetical application, since the lifetime of photodissociated Met80 is ~5-6 ps, the population of "apoptosis-active" Cyt c would be very low. Phototherapy is not the subject of this study and this sentence should be removed. (For apoptosis, analogs of cardiolipin as Cyt c ligands could be a better possibility).

We have removed this statement.

The Supplementary Informations are complete and useful.

REVIEWERS' COMMENTS

Reviewer #1 (Remarks to the Author):

The authors addressed the concerns raised for the original manuscript. In the revised manuscript, the authors improved their discussion on the kinetic model and the structural analysis. Also, the readability of the main text and SI has been improved. While I support the publication of this manuscript, the following points require further modifications.

1. The authors should cite the references more properly. For example, the citation of the reference 23 in the line 220 (“a decrease in the scattering intensity below 0.3 \AA^{-1} associated with protein thermal expansion”) does not seem appropriate. Although the low q signal was analyzed to elucidate the global structural change in the reference 23, it was argued that the change of low q signal had a non-thermal origin. Another example is the reference 36 cited in the line 221 (“The observed energy transfer to the solvent accesses the dynamics of energy transfer and equilibration between the protein and solvent”). Even though the structural change due to the solvent heating was observed in the reference, the dynamics of energy transfer is not mentioned in the reference.

2. The authors should make the statements about the simplicity of the model used for the structural analysis more clear. The authors added the lines 175 to 177 (“The structural analysis here is constrained to the first 300 fs during which the axial bonding changes significantly, and therefore changes in structure occurring at larger length scales unaddressed by our model will be of lesser importance.”). It should be mentioned that the structural change was not obtained solely from the data. Rather, the degree of structural change was assumed and with this constraint, only the heme group, His18, and Met80 were allowed to move, ignoring the structural change of the other residues in the vicinity of them.

3. With this said, this manuscript will be much strengthened if the structural change can be further justified. For example, it would be ideal if they can show that their simplified model gives better agreement with the experimental data than a control model where the structural change of other residues is also allowed.

Hytcherl Ihee

Reviewer #2 (Remarks to the Author):

The authors have more than adequately addressed my comments in the initial review. The article in my estimation is now suitable for publications in Nature Communications.

Reviewer #3 provided remarks to the Editor only, stating that they are fully satisfied with the revision and consider the paper suitable for publication in Nature Communications.

REVIEWERS' COMMENTS

Reviewer #1 (Remarks to the Author):

The authors addressed the concerns raised for the original manuscript. In the revised manuscript, the authors improved their discussion on the kinetic model and the structural analysis. Also, the readability of the main text and SI has been improved. While I support the publication of this manuscript, the following points require further modifications.

1. The authors should cite the references more properly. For example, the citation of the reference 23 in the line 220 (“a decrease in the scattering intensity below 0.3 \AA^{-1} associated with protein thermal expansion”) does not seem appropriate. Although the low q signal was analyzed to elucidate the global structural change in the reference 23, it was argued that the change of low q signal had a non-thermal origin. Another example is the reference 36 cited in the line 221 (“The observed energy transfer to the solvent accesses the dynamics of energy transfer and equilibration between the protein and solvent”). Even though the structural change due to the solvent heating was observed in the reference, the dynamics of energy transfer is not mentioned in the reference.

We have removed the comment about the origin of the difference scattering signal below 0.3 \AA^{-1} as this signal is not further analyzed or discussed in the manuscript. We have also moved reference 36 to the preceding sentence (“The transient signal prevailing beyond 10 ps exhibits the well-known change in the bulk water structure factor resulting from ultrafast energy transfer and equilibration to an elevated solvent temperature.”), where it is more appropriate.

2. The authors should make the statements about the simplicity of the model used for the structural analysis more clear. The authors added the lines 175 to 177 (“The structural analysis here is constrained to the first 300 fs during which the axial bonding changes significantly, and therefore changes in structure occurring at larger length scales unaddressed by our model will be of lesser importance.”). It should be mentioned that the structural change was not obtained solely from the data. Rather, the degree of structural change was assumed and with this constraint, only the heme group, His18, and Met80 were allowed to move, ignoring the structural change of the other residues in the vicinity of them.

To further emphasize the simplifications of our structural model in the X-ray solution scattering data analysis section, we have replaced the sentence “Starting with a ferrous cyt *c* solution structure,²⁹ we use a model for the ultrafast nuclear dynamics focused on changes in the axial ligand positions.” with “Starting with a ferrous cyt *c* solution structure,²⁹ we use a model for the ultrafast nuclear dynamics that only considers specific structural motions focused on changes in the axial ligand positions while neglecting other structural changes at the heme and global protein structural relaxation.”. The manuscript also refers to Supplementary Note 4 (Structural analysis of the sub-picosecond XSS data). We have updated this section to contain a more extended discussion of the structural modeling (see below).

3. With this said, this manuscript will be much strengthened if the structural change can be further justified. For example, it would be ideal if they can show that their simplified model gives better agreement with the experimental data than a control model where the structural change of other residues is also allowed.

To further support that the observed XSS difference signal within the first 300 fs is dominated by changes in axial ligand coordinates, we have implemented two additional models described in Supplementary Note 4. In addition to changes in Met80 and His18 positions, these models also consider distance changes between either the Cys14 or Cys17 residue and the Fe-center. For these residues which are not directly connected to the Fe-center, the photoinduced structural change associated with the photoexcitation process is more difficult to infer. Our models simply translate the Cys14 and Cys17 residues, thus changing the interatomic distances of the residues with respect to the Fe-center. While such constrained structural change does certainly not capture the subtleties of the photoinduced change at the Cys14 and Cys17 residues, we find that the inclusion of this additional degree of freedom does not significantly decrease the residual sum of squares of the fit. Moreover, we have added R² values to Supplementary Table 4. These values show that a significant part of the variation in the XSS difference signal is indeed captured by the model only considering the Met80 and His18 axial ligand positions.

Hytcherl Ihee

Reviewer #2 (Remarks to the Author):

The authors have more than adequately addressed my comments in the initial review. The article in my estimation is now suitable for publications in Nature Communications.

Reviewer #3 provided remarks to the Editor only, stating that they are fully satisfied with the revision and consider the paper suitable for publication in Nature Communications.